# From Point to Region: Accurate and Efficient Hierarchical Small Object Detection in Low-Resolution Remote Sensing Images

**Jingqian Wu** [1] **and Shibiao Xu** [2,*]

1    Department of Computer Science, Wake Forest University, Winston-Salem, NC 27019, USA; wuj18@wfu.edu
2    Institute of Automation, Chinese Academy of Sciences, Beijing 100190, China
*    Correspondence: shibiao.xu@nlpr.ia.ac.cn

**Abstract:** Accurate object detection is important in computer vision. However, detecting small objects in low-resolution images remains a challenging and elusive problem, primarily because these objects are constructed of less visual information and cannot be easily distinguished from similar background regions. To resolve this problem, we propose a Hierarchical Small Object Detection Network in low-resolution remote sensing images, named HSOD-Net. We develop a point-to-region detection paradigm by first performing a key-point prediction to obtain position hypotheses, then only later super-resolving the image and detecting the objects around those candidate positions. By postponing the object prediction to after increasing its resolution, the obtained key-points are more stable than their traditional counterparts based on early object detection with less visual information. This hierarchical approach, HSOD-Net, saves significant run-time, which makes it more suitable for practical applications such as search and rescue, and drone navigation. In comparison with the state-of-art models, HSOD-Net achieves remarkable precision in detecting small objects in low-resolution remote sensing images.

**Keywords:** small object detection; key-point prediction; image enhancement; low resolution





## 1. Introduction

Object detection plays a crucial role in image interpretation for a wide scope of applications, including intelligent monitoring, urban planning, precision agriculture, and geographic information system (GIS) updating [1]. The goal of the object detection is to identify the precise bounding box of each object in the image. In recent years, many object detection models have been proposed with high accuracy using various datasets such as COCO [2] and Pascal VOC [3]. In fact, these datasets have objects with high quality visual information since the images are in high-resolution format. Besides this, a majority of proposed solutions on different tasks are designed based on very high-resolution (VHR) data, for example: building extraction [4], tree species classification [5], and operational soil moisture mapping [6]. However, in real-world settings, it is difficult to collect high-resolution images for the objects of interest. Especially for practical remote sensing applications, aerial images can be highly resolved in terms of pixels comparing to satellite images; however, drones have a higher cost of collecting data in terms of time and energy compared to satellites. In such situations, it is very crucial to find a solution to detect small objects in low-resolution images. Small objects, defined in MS COCO [2,7], are objects that have length and width less than 32 pixels. Inspired by the above definition, considering the long slim rectangle objects in the DOTA dataset, we define small objects that have less than 900 pixels in the object bounding boxes in this paper

Many deep learning-based methods (specifically CNN based) [1] have been proposed for object detection. However, the performance of those methods on small size or low resolution is far from satisfactory (as shown in Table 1). We argue that two obvious reasons account for this problem. Firstly, resizing the input image is insufficient to distinguish the small size objects from a background (or similar categories) or achieve good localization [8].

The second reason is that the resizing operations between the layers of CNN-based methods effectively discards almost all the visual details of small sized objects, which hinders training classifiers with high accuracy. Therefore, based on these two reasons, even the state-of-the-art object detection models that have achieved impressive results on large/medium sized objects still have a poor performance on small objects in low-resolution remote sensing images (as shown in Table 1).

**Table 1.** Performance comparison of state-of-the-art object detectors on original and low-resolution remote sensing images. The detection performance decreases dramatically for small sized objects in low-resolution setting.

| Models | Original-Resolution AP | Low-Resolution AP | Decreased Value |
|---|---|---|---|
| CenterNet | 37 | 32 | 5 |
| R3detection | 51 | 39 | 12 |
| Retinanet | 26 | 23.5 | 2.5 |
| CenterNet2 | 43.1 | 28.7 | 14.4 |

To deal with the small object detection problem in low-resolution remote sensing images, we propose a general small object detection framework, which can be used by any existing object detector. In contrast to other object detection models that directly detect images, we detect a center-point as the key-point of each object which will be used, in a later stage, to pinpoint the objects. Then, we use a multi-task generative adversarial network to enhance the resolution of the image (e.g., super resolution), then detect objects (e.g., classification and bounding box regression) around the candidate positions. Our main contributions in this work are:

1.  A Hierarchical Small Object Detection Network (HSOD-Net) via The Novel Point-to-region Detection Strategy: The proposed HSOD-Net is a general small object detection framework, which can be incorporated into existing object detectors. Specifically, for low-resolution remote sensing images, we apply our key-point detector to distinguish the small sized objects from the background and get a rough estimate of the object positions; then, super-resolution is introduced to up-sample the estimated objects to a larger scale. The output from HSOD-Net can then be used as an input to existing object detectors.
2.  A Multi-task Generative Adversarial Network for Image Enhancement and Object Detection (MGAN-Det) in The HSOD-Net: There are two sub-networks in the proposed MGAN-Det, a generator network and a discriminator network. In the generator network, a super-resolution network (SRN) is provided to up-sample small blurred images into fine-scale ones and recover detailed information for more accurate detection, compared to directly re-sizing the image with bi-linear interpolation which produces less accurate images. The discriminator network tasks are: Identifying each input image patch with a real/fake score, drawing a bounding box around objects, and categorizing each detected object.
3.  The Small Object Detection Model with High Precision in Low-Resolution Remote Sensing Images: We validate the proposed HSOD-Net within a small object detection pipeline on a challenging benchmark, where our model achieves higher performance than several previous state-of-the-art models.

## 2. Related Work

In this section, we provide an overview of the most relevant work to our proposed framework HSOD-Net, including multi-scale object detection, generative adversarial network-based image super-resolution, and object detection benchmarks.

### 2.1. Multi-Scale Object Detection

In recent years, a large number of methods have been proposed for object detection, and there are two main categories of object detection: Anchor-based methods and anchor-free methods. Anchor-based methods regard the object detection problem as a regression and classification around the object area, whereas anchor-free methods do not depend on the anchor area.

There are two strategies for anchor-based methods: One-stage and two-stage detection [8]. Two-stages detection methods first generate region proposals of different objects, and then perform the regression on the bounding box and classification on the object category. Typically, two-stage object detection models consist of R-CNN [9] or fast R-CNN [10]. One stage detection methods, on the other hand, use the CNN network to predict the bounding box and object category directly, such as in Yolo [11], and SSD [12]. While two-stage detection methods normally provide more accurate detection results than that of one stage models, due to the complexity of network, they are generally slower. In contrast, anchor-free methods (e.g., CornerNet [13] and CenterNet [14]) detect key points of objects and extend to bounding box from key points, which can avoid tuning anchor related parameters. Moreover, these kind of methods avoid calculating the IoU (Intersection over Union) between ground truth bounding boxes and anchor boxes, which greatly saves time and memory resources during training.

Despite the fact that different object detection methods have achieved high quality results with different considerations and requirements, small objects detection (SOD) has always been a challenging task in multi-scale object detection [8]. Most of the above methods show unsatisfactory performance on small sized objects, since they do not have any explicit strategy to deal with such small objects. In these methods, the average precision (AP) is lower in small objects compared to large/medium sized objects. Several strategies have been developed to enhance the performance of small sized object detection. The first common strategy is multi-scaling. A typical illustration is the last layer of the SSD network [12], which combines the previously generated feature maps to form multi-scale feature maps due to a different stride, and the lower layer is normally used to detect small sized objects. However, these lower layers are limited and usually bring missing detection. Another common strategy is FPN (Feature Pyramid Network) [15]. Starting from the top layer, each lower layer is generated by up-sampling the previous layer so that the model can have feature maps in different scales. However, simply using linear interpolation to up-scale a feature map does not guarantee the effective information will be delivered. Furthermore, there is no guarantee that a particular number of layers in the pyramid is enough for a particular task, especially for small sized object detection. Thus, in this paper, we present a hierarchical small object detection network (HSOD-Net) via the novel point-to-region detection strategy.

### 2.2. Generative Adversarial Network Based Image Super-Resolution

Image super-resolution (SR), without any prior information, is an ill-conditioned problem. With enough training data, CNN-based methods have recently achieved great progress in this problem. For instance, SPSR [16] encodes a sparse representation prior into their feed-forward network architecture based on the learned iterative shrinkage and thresholding algorithm. DRCN [17] uses a recursive structure to decrease the number of network parameters. In these solutions, the optimization goal is often to minimize the mean squared error (MSE) between the recovered high-resolution (HR) image and the ground truth image. Minimizing MSE also maximizes the peak signal-to-noise ratio (PSNR), which is another common measure used to evaluate and compare SR algorithms. However, the ability of MSE and PSNR to capture perceptually relevant differences is very limited, as they are defined based on pixel-wise image differences. As a result, very high PSNR does not necessarily reflect perceptually a good SR result.

SRGAN [18], Pix2Pix [19] and Pix2PixHD [20] tackle the super-resolution problem by employing a generative adversarial network (GAN). GAN [21] was first proposed

by Goodfellow in 2014, which is a model containing two networks, a generator and a discriminator. The generator generates images based on the learned features, and the discriminator identifies whether the generated image is real or generated by the generator. The purpose is to improve the performance of both generator and discriminator through their adversarial relationship. As a state-of-the-art GAN-based super-resolution method, SRGAN [18] provides a powerful framework for generating plausible-looking natural images with high perceptual quality. SRGAN encourages the reconstructions to move towards regions of the search space with a high probability of containing photo-realistic images and thus closer to the natural image manifold. Building on SRGAN's architecture and to deal with the small object detection problem in low-resolution remote sensing images, we propose a multi-task generative adversarial network for image enhancement and object detection (MGAN-Det).

### 2.3. Object Detection Benchmarks

Recently, a large number of datasets have been built for object detection which can be separated into two types. The first type contains datasets of natural scene images, such as the PASCAL VOC dataset [3] and the COCO object detection dataset [2]. Images in these datasets are typically of high quality with the objects of interest shown in relatively large scale. The second type of dataset for object detection has datasets representative of optical remote sensing images, such as DOTA [22], DIOR [1], and RSOD [23]. In contrast to the first dataset type, these datasets contain images with objects at relatively low spatial resolutions, which is characteristic of remote sensing imagery.

Detecting small objects (objects that have a relatively small size in an image) in low-resolution image is challenging because small objects usually lack sufficient detailed appearance information, which can distinguish them from the background or similar objects. To solve this problem, we propose a general small object detection framework in low-resolution remote sensing images, which can be easily incorporated into existing object detectors (e.g. CenterNet [8], CenterNet2 [24], Retinanet [25] or R3detection [26]). Moreover, we experimentally validate our proposed framework on two public detection benchmarks for small objects, DOTA [22] and COCO [2].

In this paper, we argue that these state-of-art object detection models (e.g., CenterNet [8], CenterNet2 [24], Retinanet [25] and R3detection [26]) work well on the remote sensing benchmarks in their original resolution, but the detection performance, especially for small sized objects in low resolution (e.g., four times down sampled image as the low-resolution representation), would decrease dramatically by a large margin, as shown in Table 1.

### 3. Methodology

In this section, we introduce our small object detection framework, HSOD-Net, which can be easily embedded into existing object detectors. There are two main stages via a hierarchical point-to-region detection strategy. In the first stage, a key-point detector and an embedded data processor are used to distinguish the small sized objects from the background. In this stage, we take the low-resolution image as input, and extract key-point guided rough object positions. This technique is more robust than the direct object detection which requires more visual details. In the second stage, the regions surrounding the candidate positions are fed into a multi-task generative adversarial network (MGAN-Det) to enhance images (e.g., super resolution) and then detect objects (e.g., classification and bounding box regression) from the resolved image. The overall architecture of our proposed HSOD-Net is shown in Figure 1 and each stage is explained in the following subsections.

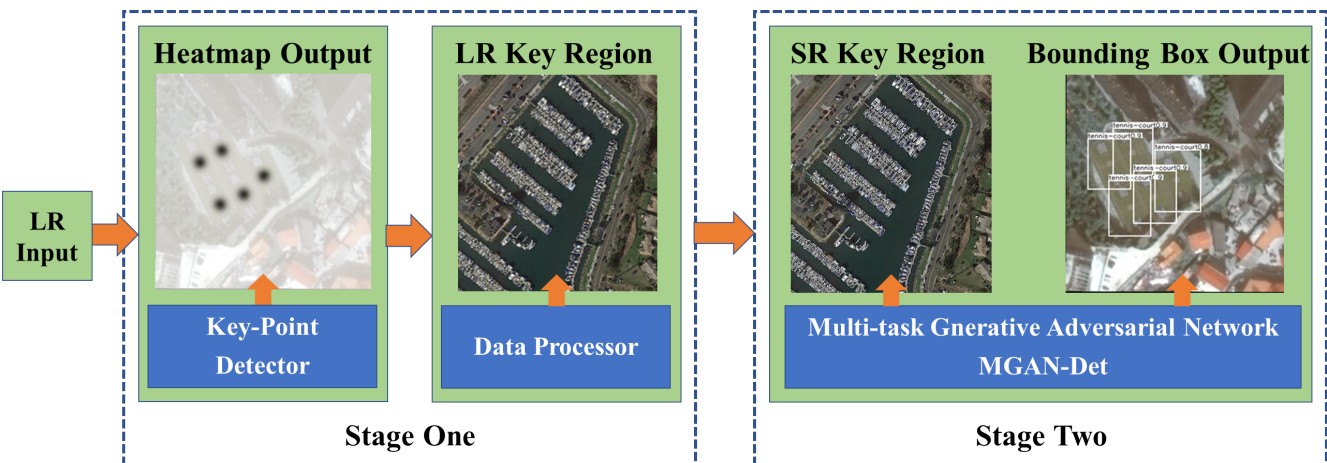

**Figure 1.** The architecture of HSOD-Net. It has two stages for our hierarchical point-to-region detection strategy, including key-point guided rough object position extraction and multi-task GAN.

### 3.1. Key-Points Guided Rough Object Positions Extraction

Due to the excessive down-sampling operations in CNN-based methods, the learned feature maps are insufficient for detecting the small sized objects and training a high-quality classification network. To overcome this problem, we propose a key-points guided rough object positions extraction module, motivated by the idea that identifying the center-point as the key-point of each small sized object is more robust than creating bounding boxes around the objects. This module consists of two parts: The key-point detector and the embedded data processor.

#### 3.1.1. Key-Point Detector for Small-Sized Objects

Inspired by [14], we develop a general network architecture for center-point detection as shown in Figure 2. Let $I \in \mathbb{R}^{W \times H \times 3}$ represent the input low-resolution image with three channels in color space, and set $W = H = 512$ as the size of each image. The Hourglass network [13,27] is deployed to extract a feature map of $128 \times 128 \times 256$ from the input image. Then, two parallel processes are executed to calculate the key point of objects and the center offset. The center-point is regarded as the only key-point of the the final output, which is a heatmap represented as $\hat{H} \in [0,1]^{\frac{W}{R} \times \frac{H}{R} \times C}$, where $R$ is a default output stride and $C$ is the number of categories. The key point detector will predict all possible vital points, with a probability greater than or equal to a threshold value. (The threshold value is set to be 0.15 for all experiments empirically). For example, the key point detector predicts five vital points in stage one of Figure 1.

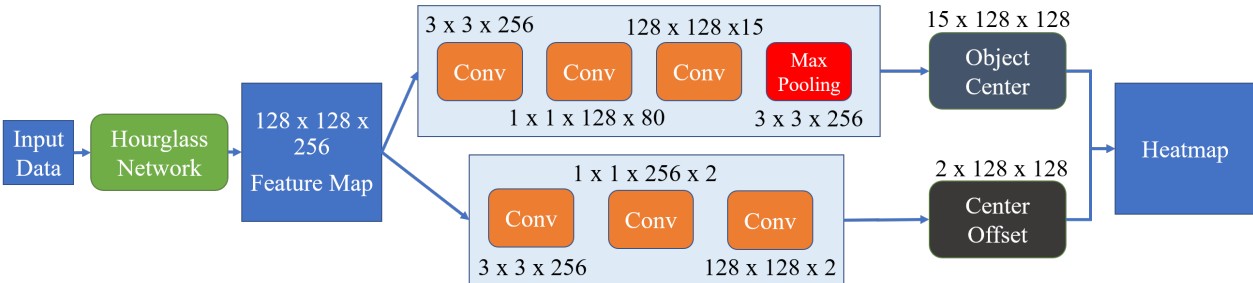

**Figure 2.** The network architecture for the key-point detector.

The key-point detection network uses two loss functions: Key-point loss and offset loss. The key-point loss denoted as $L_k$ reflects whether the model can pinpoint the key-

point to the center of the object, which is a pixel-wise logistic regression with focal loss defined as follows [25]:

$$L_k = \frac{-1}{N} \sum_{xyz} \begin{cases} ((1 - \hat{Y}_{xyz})^{\alpha} log(\hat{Y}_{xyx}) & if \ \hat{Y}_{xyz} = 1 \\ (1 - \hat{Y}_{xyz})^{\beta}(Y_{xyz})^{\alpha} log(1 - \hat{Y}_{xyz}) & otherwise \end{cases} \tag{1}$$

where $\alpha$ and $\beta$ are defined as hyper-parameters of the focal loss; we set $\alpha = 2$ and $\beta = 4$ in our experiments. For each ground truth key point $p \in \mathbb{R}^2$, a low-resolution (i.e., down-sampled) equivalent $\hat{p} = \lfloor \frac{p}{R} \rfloor$ is presented. Every single key point is squeezed into a separate heatmap $Y \in [0, 1]^{\frac{W}{R} \times \frac{H}{R} \times C}$, using a Gaussian kernel $Y_{xyz} = exp(-\frac{(x - \hat{p}_x)^2 + (y - \hat{p}_y)^2}{2\delta^2 p})$, where $\delta_p$ is an object size-adaptive standard deviation, and $N$ is the number of key-points in the image, which is used to normalize all positive focal loss instances to 1.

The offset loss $L_{off}$ reflects the shift in all pixels during the downsampling process caused by the output stride $R$. A local offset $\hat{O} \in R^{\frac{W}{R} \times \frac{H}{R} \times 2}$ is defined for each center point as follow:

$$L_{off} = \frac{1}{N} \sum_p |\hat{O}_{\hat{p}} - (\frac{p}{R} - \hat{p})| \tag{2}$$

The total detection loss, denoted $L_{key}$, is defined:

$$L_{key} = L_k + \lambda_{off} L_{off} \tag{3}$$

where $\lambda_{off}$ is set to be 1 in our experiments, so the effects of these two loss functions are equivalent.

### 3.1.2. Embedded Data Processor for Estimating Object Regions

After the heatmap based key-points are extracted from the key point detector, the data processor is deployed to predict the regions around the existing key-points. In this paper, we use the K-means algorithm to fit $k$ key-point predictions into $n$ regions based on the Euclidean distance. The data processing includes the following steps:

1. Randomly select $n$ key-points among all detected key-points as the clustering centers of the initial regions.
2. For the other detected key points, calculate the Euclidean distance to these given clustering centers, and mark the closeted key-points as parts of one region.
3. Readjust the current region center when a new key-point is added into that region.
4. If all the region centers do not move, then the algorithm converges. Otherwise, we repeat step 2.

By using this simple and effective technique, we cluster the detected $k$ key points into $n$ regions. Figure 3 shows an example of how regions are formed from key points. The red boxes are the ground truth bounding box annotations, shown in Figure 3a to reveal the ground truth objects. The green dots presented in sub-image Figure 3a are the detected key points generated from our key-point detector, and the yellow boxes presented in Figure 3b are the round regions extracted by our data processor based on the predicted key-points location in Figure 3a. Figure 3c displays the extracted regions by the data processor, which will be used in later stage of HSOD-Net.

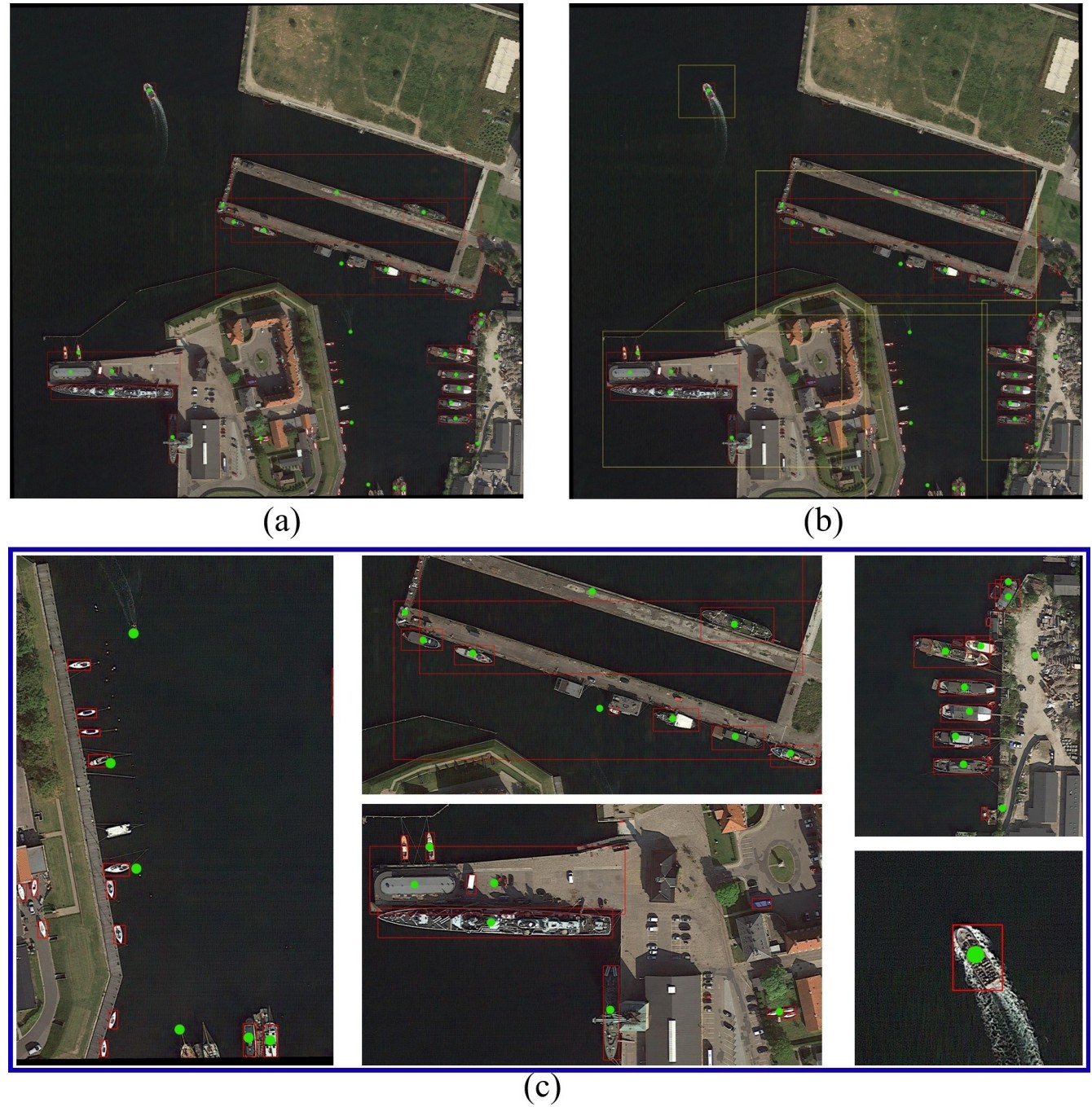

**Figure 3.** Typical example of key-point guided rough object position extraction. The red boxes are the ground truth bounding box annotations, while the green dots are the detected key points generated from our key-point detector, and the yellow boxes are the round regions extracted by our data processor.

The usage of this method significantly improves the efficiency of our model. In Figure 3, this typical image from the DOTA dataset has the resolution of $4654 \times 4697$. Traditional slider based methods create batch images to train and test super-resolution models. For instance, using a slider of size $1000 \times 1000$ would extract $9 \times 9 = 81$ images. Through our proposed approach, only five regions will be extracted from each image and fed into the next stage. Therefore, efficiency is dramatically improved.

### 3.2. Multi-Task Generative Adversarial Network for Image Enhancement and Object Detection

Inspired by [18,28] and, we introduce our multi-task Generative Adversarial Network (MGAN-Det) to super resolve a given low-resolution image, and detect up-sampling objects in this section. There are two sub-networks in the proposed MGAN-Det, a generator network and a discriminator network; their network architectures are shown in Figures 4 and 5.

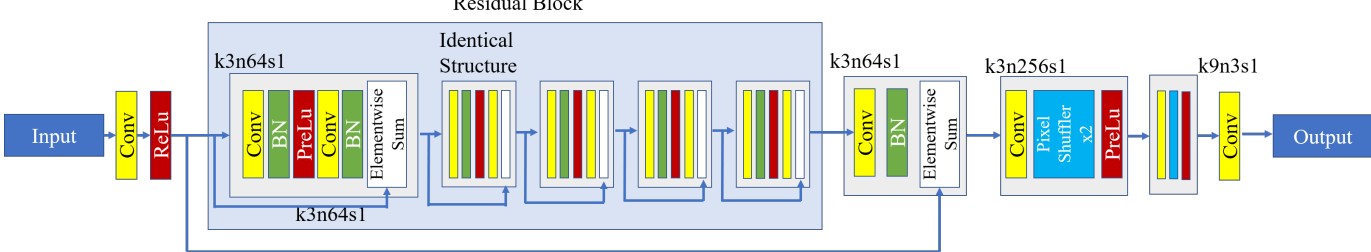

**Figure 4.** Generator Model Structure: K stands for kernel size, n stands for number of feature map, and s stands for strides.

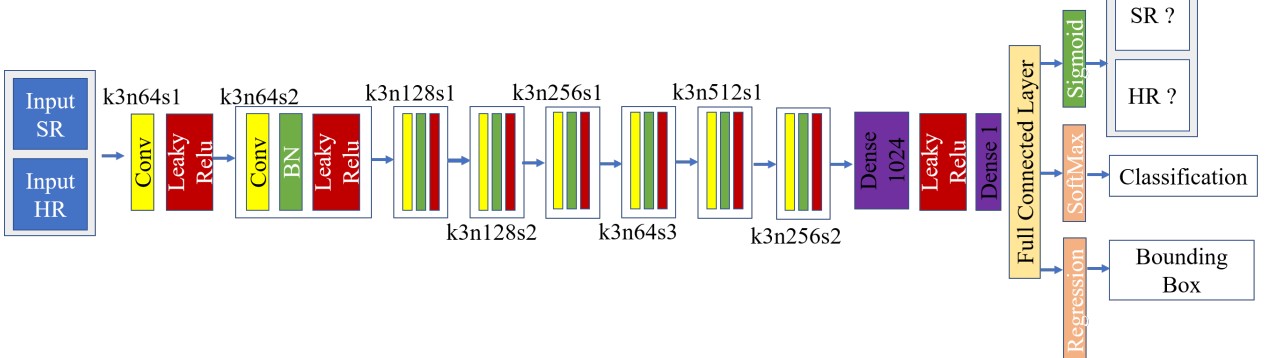

**Figure 5.** Discriminator Model Structure: For example, k3n64s1 stands for a conv layer with kernel size of 3, 64 feature map, and a stride of 1.

For the super-resolution generator network, a low-resolution image $I^{LR}$ is generated by down-sampling a high-resolution image $I^{HR}$ with a scale factor $r$. Let the size of $I^{LR}$ be $W \times H \times C$, our goal is to estimate its high-resolution equivalent by producing a super resolved image $I^{SR}$ with size of $rW \times rH \times C$. Then, the output of the generator network (super-resolved images) is used by the discriminator to classify as a fake generated super-resolved (SR) image or an original high-resolution (HR) image and to perform object detection (object classification and bounding-box regression), respectively.

For the proposed multi-task problem, our MGAN-Det model adopts the content loss and adversarial loss for super-resolution, while the object classification and bounding box regression use the softmax and smooth $L1$ loss functions after each full connected layer. To compute the content loss function, the SR image and the HR image are fed into VGG19 network [29] to extract their features. The content loss is calculated by the Mean Square Error (MSE) of these two extracted feature maps as follows:

$$loss_1 = \frac{1}{N} \sum_{i=1}^{N} (\phi((I_i^{HR})) - \phi(G_\theta(I_i^{LR})))^2 \tag{4}$$

where $\phi$ is the feature map. The content loss represents the Euclidean distance between the reconstructed feature map $G_\theta(I^{LR})$ and the ground truth feature map $I^{HR}$. The adversarial

loss is set to fool the discriminator network into classifying the generated SR images as HR images, which is defined as follows:

$$loss_2 = \sum_{i=1}^{N} -logD_\omega(G_\theta(I_i^{LR}))$$ (5)

where $D_\omega(G_\theta(I^{LR}))$ is simply the probability that the reconstructed image $G_{\theta G}(I^{LR})$ is the ground truth HR image.

To classify the object category, we use a classification loss function. We denote $I_i^{LR}$ and $I_i^{HR}$ to be the LR image representation and the ground truth HR image presentation; i ranges from 1 to N, which covers all objects. Denote $u_i$ as the corresponding category, and $u_i$ should be in the range of 1 to k, where k is the number of the categories of objects. $D_\delta(G_W(I_i^{LR}))$ is the probability that the generated SR image belongs to the true category $u_i$, and $log(D_\delta(I_i^{HR}))$ is the probability that the ground truth HR image belongs to the true category. The classification loss is defined as follows:

$$loss_3 = \frac{1}{N} \sum_{i=1}^{N} -(log(D_\delta(G_\omega(I_i^{LR})))) + (log(D_\delta(I_i^{HR})))$$ (6)

Furthermore, a regression loss is used to localize the final bounding boxes around the objects, which is defined as follows:

$$loss_4(t,v) = \frac{1}{N} \sum_{i=1}^{N} \sum_{i \in x,y,w,h} \mathbf{u}_i(smooth_{L1}(t_{i,j}^{HR} - u_{i,j}) + smooth_{L1}(t_{i,j}^{SR} - v_{i,j}))$$ (7)

$$smooth_{L1}(x) = \begin{cases} 0.5x^2, & if|x| < 1 \\ |x| - 0.5, & otherwise \end{cases}$$ (8)

where $\mathbf{u}_i$ represents an indicator, $v_i$ denotes the tuple of the bounding box regression target (x, y, w, h), and $t_i$ denotes the predicted regression tuple in the same format. $t_i^{HR}$ and $t_i^{SR}$ are the *i*-th bounding box tuple of the ground truth HR image, and the generated SR image.

Finally, we combine the above loss functions to train the generator and discriminator networks together, and optimize the parameters of generator *G* while keeping the parameters of discriminator *D* fixed, and vice versa. Therefore, the two networks are iteratively optimized.

### 3.3. Embedding HSOD-Net with Existing Detectors for Small-Sized Object Detection

HSOD-Net, as a multi-task model, also allow incorporation of other baseline detectors. In order to incorporate our proposed framework HSOD-Net into existing object detectors, we construct it into three parts, including key-point detector, super-resolution image generator, and an existing object detector, which is connected last. We select four state-of-the-art object detectors: CenterNet [8], CenterNet2 [24], Retinanet [25] and R3detection [26]. Retinanet and two versions of CenterNet are known for their innovative model structure, and they have excellent performance on COCO datasets and Pascal VOC datasets. R3detection is a newly proposed model that mainly focuses on DOTA datasets. These detectors can be attached to our HSOD-Net, making full use of its generated super-resolution images. According to Table 2, each of these baseline models performs poorly on low-resolution DOTA datasets. The lack of resolution leads to a lack of information to detect ground truth objects. However, our HSOD-Net improves the detection results, and different baseline models have different degrees of improvement.

**Table 2.** Baseline Model Performance and Our Model Embedded Performance for Small Object Detection in Low-Resolution Remote Sensing Images.

| Baseline Models | HR AP | LR AP | Decreased Value | Our Model Embedded LR AP | Improved Value |
|---|---|---|---|---|---|
| CenterNet [8] | 37 | 32 | 5 | 36.7 | 4.7 |
| R3detection [26] | 51 | 39 | 12 | 49.0 | 10 |
| Retinanet [25] | 26 | 23.5 | 2.5 | 25 | 1.5 |
| CenterNet2 [24] | 43.1 | 28.7 | 14.4 | 39.5 | 10.8 |

## 4. Experiments

In this section, we present our detailed experiments. First, we use the DOTA dataset to provide: A quantitative analysis of different detection methods for small object in remote sensing images, an ablation study for HSOD-Net, and visual comparisons of different detection methods. Second, the evaluation on COCO dataset is presented to prove the robustness of HSOD-Net for small object detection in nature scene images. Finally, the time complexity of the proposed HSOD-Net is studied to prove the advantage of our HSOD-Net efficiency.

### 4.1. Evaluation on DOTA Dataset for Small Object Detection in Remote Sensing Images

In this paper, we use DOTA [22] dataset which has 2806 remote sensing images to evaluate the performance of our proposed HSOD-Net. Each image in the DOTA dataset has a size ranging from 800–5000 pixels in width and height. For images larger than 2000 pixels, directly using the existing state-of-the-art object detection methods is not feasible since they require the size of the input image to be either 300, 512, or 608. Without pre-processing for the large images, those detection methods will resize the images automatically, which will lead to poor training and testing. Therefore, we clean the dataset using a slider so large images are cut to 1024 × 1024 with a step of size 512 to satisfy the size requirement while preserving sufficient information in the images.

#### 4.1.1. Quantitative Comparison with Low-Resolution Remote Sensing Images

The state-of-the-art object detection models (e.g., CenterNet [8], CenterNet2 [24], Retinanet [25], and R3detection [26]) achieve high performance on images with high resolutions. Here, we study the performance of those methods on detecting small sized objects in low resolution (e.g., four times down sampled image as the low-resolution representation). In Table 3, we provide the average precision (AP) performance comparison for detecting objects in low-resolution inputs, and our HSOD-Net has achieved a remarkable performance. It is worth noting that R3detection works slightly better than our method, that is because we only take the VGG19 network as our backbone, which is simple and effective. Moreover, as stated in Section 3.3, our proposed HSOD-Net can be incorporated into these existing object detectors to improve the detection results, and while different baseline models have different degrees of improvement, R3detection achieves the best performance(as Table 2 shows).

**Table 3.** Model performance on low-resolution DOTA dataset.

| Performance and Methods | Our HSOD-Net | CenterNet | CenterNet 2 | Retinanet | R3detection |
|---|---|---|---|---|---|
| Low-Resolution AP | 37.6 | 32 | 28.7 | 23.5 | 39 |

To evaluate the super-resolution task of HSOD-Net, Figure 6 shows the visual comparison of the original image, the super-resolved image using HSOD-Net, the down-sampled image by scale of four, and the bi-linear interpolation based up-sampling image. Both our visual image quality and quantitative PSNR are better than that of bi-linear interpolation for a random sample from the DOTA validation dataset.

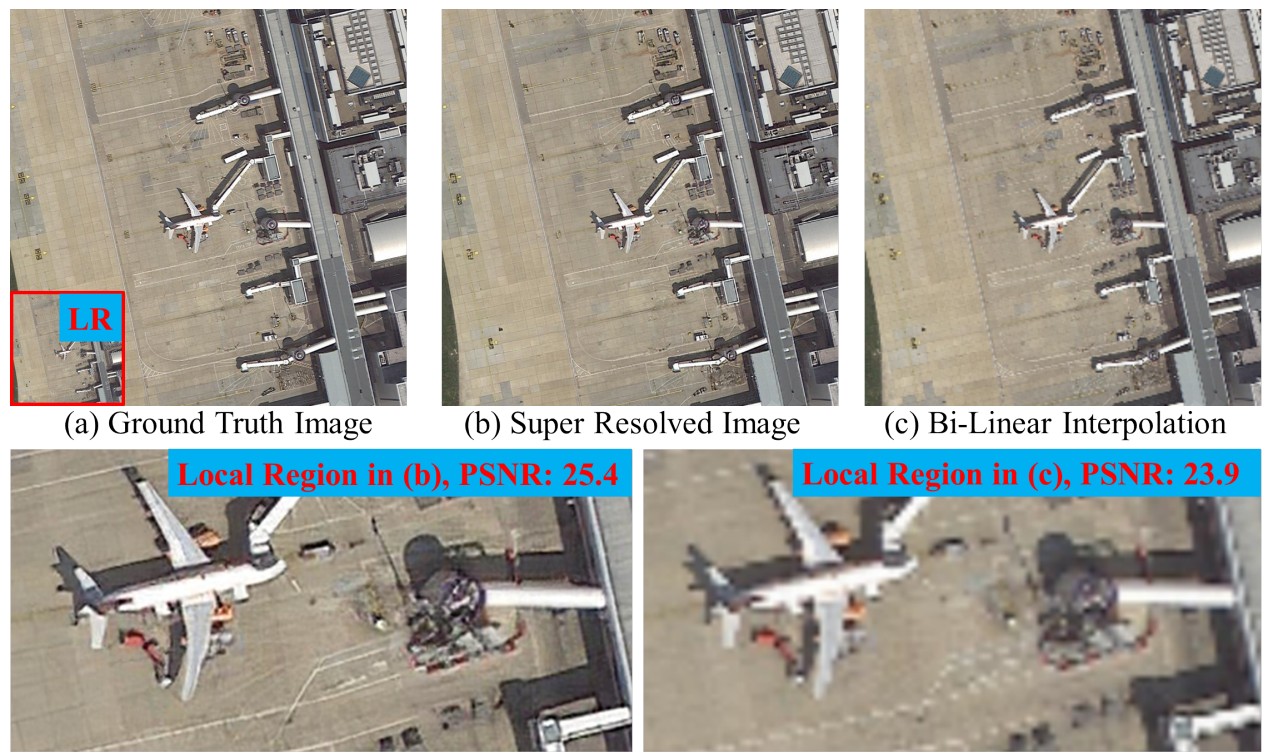

(a) Ground Truth Image     (b) Super Resolved Image     (c) Bi-Linear Interpolation

(d) Visual Comparison of Local Region

**Figure 6.** Visual comparison of two up-sampling methods (our HSOD-Net and bi-linear interpolation) for two random images in DOTA validation set.

### 4.1.2. Ablation Experiments on Key-Point Detection

In HSOD-Net, we use the key-point detection model to generate a heat map representing the distribution of probabilities for potential objects of an image. As in all heat maps, the inner part of the object has higher probability than its edges and the local maximum probability stands for the center point of the object. The predicted center points are often correlated with the higher probabilities which allows us to estimate the rough locations of the objects. With this key-point detection structure, the model predicts the top $K$ probabilities as potential objects. In order to determine the ideal $p$ ($p$ is the threshold that among all top K potential objects, only those that have probability greater than p are kept), we develop an evaluation method called hit-accuracy $H$. The idea of the term "hit" means the accuracy of the predicted center point located in the ground truth object bounding box area. The key-point detection model gives a certain number of predictions with probabilities greater than $p$. Among these predictions, $M$ is the number of hit objects. $N$ is the number of ground truth objects. Therefore, we define hit-accuracy as $H = M/N$. In our experiment, we empirically set $p = 0.15$ to train with low-resolution DOTA dataset.

In this section, we perform an ablation study to pinpoint which parts of our proposed framework contribute the most to the overall performance. We consider the key-point detection model and the super resolution via MGAV-Det model during testing as the comparison metrics. To measure the effectiveness of the key-point detection, we perform two tests; the first one is done by applying the super resolution on the entire images, then the baseline object detection model (CenterNet [8]) is used to detect the objects on the super-resolved images because it is a state-of-art detection model, and the other object detector will give similar results. The second one is done by using our proposed method of applying the key-point detection model to generate regions around the extracted points. Those regions are then super-resolved individually before applying the object detection model. The AP scores of both tests are compared to illustrate the effectiveness of the key-point detection method (Table 4). The hit accuracy is also compared to measure the

degree of the missed objects by the detector. Furthermore, the inference time is compared to illustrate the efficient of our key-point guided strategy.

**Table 4.** Ablation Experiments on Key-point Detection.

| Method | Low-Resolution AP | Hit Accuracy | Inference Time |
|---|---|---|---|
| Key point detection embedded (CenterNet as baseline) | 36.7 | 0.757 | 0.27 s/task |
| No key point detection (CenterNet as baseline) | 35.4 | 0.74 | 0.77 s/task |

As illustrated in Table 4, key-point guided strategy indeed improves the detection accuracy (e.g., AP score) for low-resolution images. While it is very common to miss detecting small objects on low-resolution images, the hit accuracy with a key point detector is slightly improved over the other test. Moreover, the embedded key point detection improves the inference time per task by 0.5 seconds. By using a key point detection, the regions of the potential object are proposed, and we only need to super resolve those regions and detect those regions, which makes our approach more efficient.

### 4.1.3. Ablation Experiments on Super-Resolution

Similar to the previous ablation study, two tests are set up to examine the impact of the super resolution. The first test consists of a key-point detector to propose a set of rough regions, and CenterNet, as the object detection model, is used to detect objects on those potential regions. The second test is our proposed method with the super resolution via MGAV-Det model and using the same object detection model. The AP score is compared to illustrate the effects of the super-resolution stage in detecting low-resolution data. Table 5 shows a significant performance improvement with a super-resolution embedded model.

**Table 5.** Ablation Experiments on Super Resolution.

| Method (CenterNet as Baseline) | No Super-Resolution Embedded Model | Super-Resolution Embedded Model |
|---|---|---|
| Low-Resolution AP | 18.8 | 36.7 |

### 4.1.4. Visual Comparison with Low-Resolution Remote Sensing Images

In this section, we provide a visual comparison of predicted bounding box on randomly selected low-resolution remote sensing images from the DOTA dataset: P2789 as scene 1 and P0331 as scene 2. In Figure 7a,b, the ground truth bounding boxes are shown first, followed by the prediction of HSOD-Net, and then the predictions from the different baseline models. The score of the bounding box is also shown in the figures. We only show bounding boxes that have scores above 0.8 out of 1. The results show the advantages of our proposed method in detecting small objects (e.g., small sized cars and boats in the figures) in low-resolution remote sensing images.

### 4.2. Evaluation on COCO 2014 Dataset

Though our main problem setting focuses on small object detection in low-resolution remote sensing, HSOD-net can be also deployed on other natural scene datasets. We test the robustness of HSOD-net on COCO 2014 datasets. HSOD-Net is trained on the COCO 2014 train set, and the tested on COCO 2014 val set. CenterNet as a baseline model is embedded in our framework to validate the improved detection performance on low-resolution natural images (as Table 6 shown).

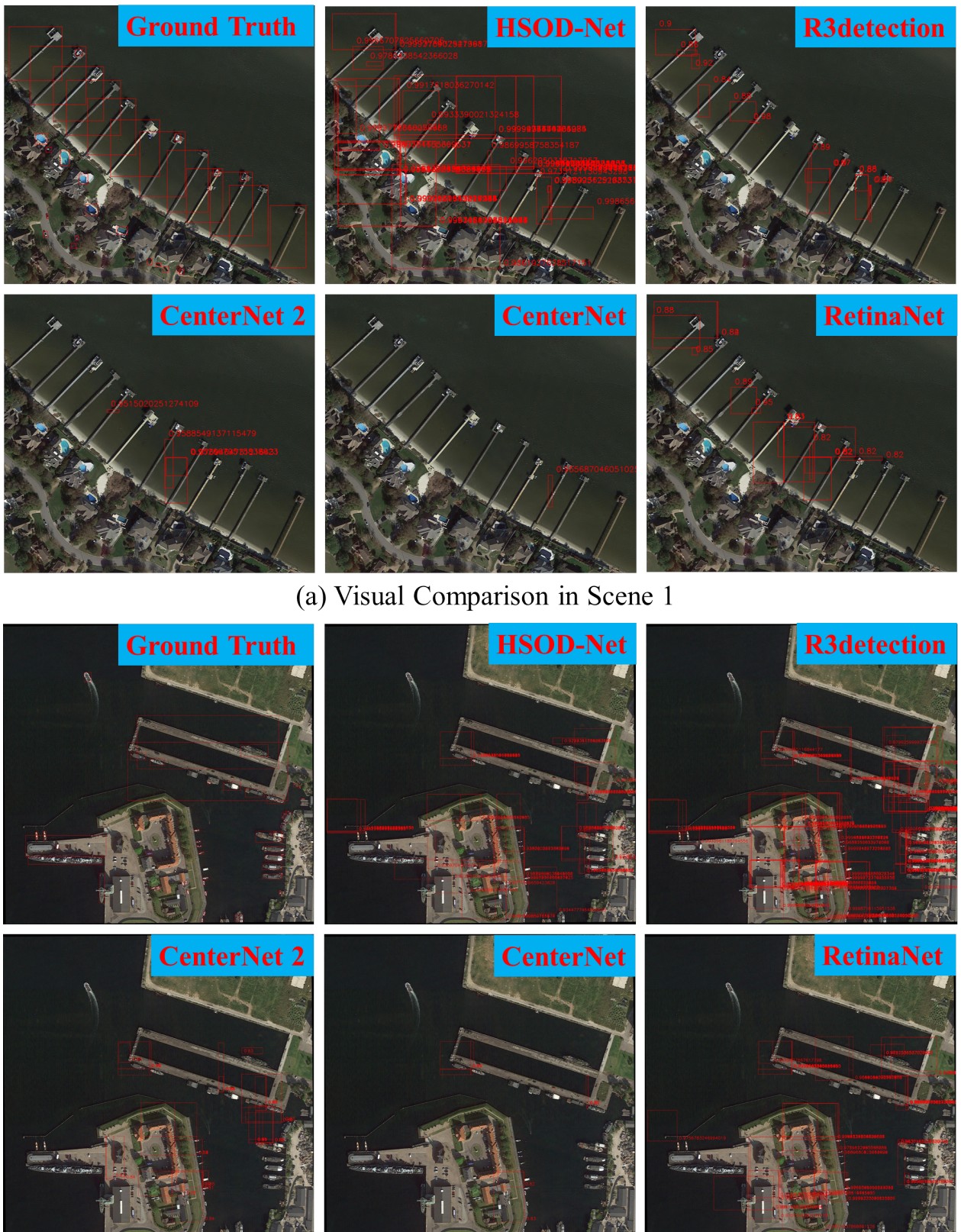

**Figure 7.** Visual comparison of different detection methods for two randomly selected image from the DOTA validation set.

**Table 6.** Quantitative Comparison on COCO 2014 Dataset.

| | w/o Embedded on HSOD-Net | Embedded on HSOD-Net | Improvement |
|---|---|---|---|
| CenterNet as baseline | 19.4 | 28.0 | 8.6 |

*4.3. Computational Efficiency Analysis and Comparison*

As discussed above, efficiency is a very important factor for evaluating deep learning models. For instance, using the key point detection strategy reduces the time complexity as shown in Table 4. Deep learning models for super resolution and object detection require training and testing on large images in patches cut by a slider. The problem with this approach is that extensive computation repetitions occur since the step is usually half the size of the slider, and the overlapping areas are calculated several times. Moreover, the background of a remote sensing image takes up a large proportion of the entire image which wastes the computation time on areas of no interest. In contrast, our proposed HSOD-Net eliminates this problem by embedding a key point detector at the beginning. After the regions pointed by the key point detector are known, other areas are no longer considered, which drastically reduce the computation time. Table 7 contrasts the improved inference time between two kinds of methods on different baseline models, which are traditional super resolution with object detection on whole image, and our proposed method embedded with different detection models.

**Table 7.** Inference Time Comparison.

| Detection Models | Traditional Method Speed | Proposed Method Embedded Speed | Improved Value |
|---|---|---|---|
| CenterNet | 0.77 s/task | 0.27 s/task | 0.5 s/task |
| R3detection | 0.94 s/task | 0.59 s/task | 0.35 s/task |
| Retinanet | 0.93 s/task | 0.57 s/task | 0.36 s/task |
| CenterNet2 | 0.75 s/task | 0.21 s/task | 0.54 s/task |

**5. Conclusions**

In this paper, we propose a hierarchical small object detection network (HSOD-Net) via the novel point-to-region detection strategy. For low-resolution remote sensing images, the key-point detector is applied to distinguish the small size objects from the background and estimate the object positions; then, super-resolution is used to up-sample a small object image into a larger scale required by the existing object detectors. Furthermore, we provide a multi-task generative adversarial network for image enhancement and object detection (MGAN-Det) in the HSOD-Net. We validate the proposed framework within the small object detection pipeline on a challenging benchmark, where our detection model achieves higher performance than several previous state-of-the-art models. The hierarchical strategy is also proven to be more efficient in terms of running time, which is suitable for practical application.

The proposed model can be applied to small object detection in the satellite photography area. Images captured from a satellite have relatively lower resolution compared to images collected by drones or surveillance camera. Furthermore, objects in a satellite image have relatively smaller size. Traditional object detection models bring misdetection and inaccurate bounding boxes due to lack of resolution information and small sized objects in satellite data. However, the proposed method specializes in detection of small objects in low-resolution images. More specific application could include real-time satellite monitoring, land-scape exploration via satellite and so on.

In this paper, we mainly focus on small sized object detection on remote sensing images; there will be no movement of the identified object in the still image. Therefore, the effectiveness and accuracy of the proposed method are maintained and not influenced by the dynamic objects of the input image. In the future, the proposed method can be

expanded to process video inputs. Then, the object detection coherence of serialized images and the efficiency of multi-frame processing should be further considered, which may be solved through optical flow estimation for key frames.

**Author Contributions:** Investigation, J.W.; Methodology, J.W. and S.X.; Software, J.W.; Supervision, S.X. All authors have read and agreed to the published version of the manuscript.

**Funding:** This research received no external funding.

**Conflicts of Interest:** The authors declare no conflict of interest.

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
