# Peer review of "From Point to Region: Accurate and Efficient Hierarchical Small Object Detection in Low-Resolution Remote Sensing Images"

_remotesensing, doi:10.3390/rs13132620_

Round 1
Reviewer 1 Report
The original scientific achievement of the authors of the paper is the development of the following tasks: Hierarchical Small Object Detection Network (HSOD-Net) via The Novel Point-to-region Detection Strategy, Multi-task Generative Adversarial Network for Image Enhancement and Object Detection (MGAN-Det) in The HSOD-Net and Small Object Detection Model with High Precision in Low Resolution Remote Sensing Images.
Comments:
1. What will the effectiveness and accuracy of the described method be influenced by the movement of the identified object, such as a ship or an airplane?
2. Conclusions is only a summary of the paper. There are no specific conclusions from the research carried out here.
3. In Conclusions, provide the direction of further research and application of the described method.
Author Response
Dear reviewer,
Thank you for your kindly and valuable comments. We have prepared a response word document attached below.
In addition, we have updated our paper and highlighted the changes.
Thank you again for your time and effort!

Reviewer 2 Report
The authors present a method for detecting small objects based on a multi-stage pipeline. First, they use a neural network to pinpoint candidate object centers, then they use a second network which simultaneously improve the resolution of the candidate areas and perform final object detection.
The approach appears to work well on the dataset provided and the authors discuss improvements in term of accuracy and execution time with respect to other methods.
I feel that the term "small objects" need to be clarified. There is no definition on when an object should considered "small". Is it a matter of resolution or number of pixels? A more quantitative discussion of the subject would be beneficial.
Regarding section 3.1.1, I found it difficult to follow. Unordered questions follows. While the answer might be straightforward, maybe the authors could rephrase the text accordingly.
1) In Figure 2 we have the network structure but not the dimension of the outputs (center point and offset). I didn't manage to decide if the network proposes all the possible keypoints at once, which are then merged in a single heatmap, or not. However, the authors say that "the center point is regarded as the only key point of the final output". This is not clear to me especially looking at the heatmap in Figure 1. Otherwise, if the network find only one key-point per image, how that heatmap in Figure 1 is produced? This is not discussed.
2) C is never defined. I don't get why a heatmap should have C channels.
3) If the network finds a keypoint in image coordinates, why we need to parametrize this as a center and an offset? Wouldn't only the center suffice? If not, why? The terms object center and center offset, although clearly self-explanatory for the authors, are not as such for me.
Author Response

(The authors gave the same response as above.)
